# MMUDA: Towards Robust Sleep Staging with Multi-Source Multi-Channel Unsupervised Domain Adaptation

## Abstract

The generalization of deep learning models for sleep staging across different datasets is severely hindered by domain shift, a critical obstacle for their clinical adoption. We introduce MMUDA, a novel framework that tackles the complex, real-world challenge of Multi-source Multi-channel Unsupervised Domain Adaptation. Our approach learns domain-invariant features from multiple labeled source domains and an unlabeled target domain through a carefully designed architecture. We employ dedicated encoders with channel attention to capture rich temporal context and enhance inter-channel feature fusion. To bridge the domain gap, MMUDA uniquely combines two complementary alignment strategies: Maximum Mean Discrepancy (MMD) explicitly minimizes the distribution discrepancy between domain pairs, while cross-domain contrastive learning (CL) ensures that the aligned features remain class-discriminative. This dual-alignment process is stabilized by a variational autoencoder (VAE) that encourages a more compact latent feature space. Comprehensive evaluations on several public sleep datasets show that MMUDA sets a new state-of-the-art in cross-domain sleep staging without requiring any target domain labels. Furthermore, we confirm its robustness and practical utility in locally collected hospital data. Our code will be released upon acceptance.

## 1 INTRODUCTION

Sleep staging, the process of classifying sleep into distinct phases, is fundamental to diagnosing and understanding sleep disorders that affect millions worldwide Itani et al. (2017); Finan et al. (2016). The clinical gold standard relies on expert manual scoring of polysomnography (PSG) recordings Basha et al. (2020), which capture a rich array of physiological signals, including electroencephalogram (EEG), electrooculogram (EOG), and electromyogram (EMG). This manual process, however, is exceptionally labor-intensive, time-consuming, and prone to inter-rater variability, creating a significant bottleneck for large-scale clinical application and research Danker-Hopfe et al. (2009).

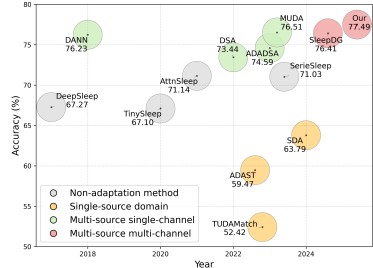

Figure 1: Comparison with baseline models. Our method achieves the highest ACC (77.49%), surpassing all baselines.

The advent of deep learning has ushered in a new era of automated sleep staging, with models demonstrating impressive, even human-level, performance in intra-dataset evaluations Loh et al. (2020). However, a critical barrier impedes their translation to real-world clinical practice: the problem of **domain shift** Pooch et al. (2020). When a model trained on data from one source (e.g., a specific hospital or device) is deployed on data from another, its performance often degrades catastrophically Fan et al. (2022).This brittleness stems from inevitable variations in data acquisition protocols, hardware, and patient populations, rendering most existing models unreliable for widespread clinical use Attia et al. (2024).

To mitigate this challenge, researchers have turned to Unsupervised Domain Adaptation (UDA) Liu et al. (2022), which aims to adapt models to unlabeled target domains. Yet, current UDA methodologies for sleep staging suffer from two critical limitations that overlook the complexities of clinical data. First, they predominantly operate under a simplified single-source Shi et al. (2024), single-target assumption, failing to address the reality that clinical data is often aggregated from **multiple, heterogeneous sources**. Applying single-source techniques in such scenarios can be ineffective or even detrimental due to conflicting domain characteristics. Second, many approaches do not adequately model the synergistic and complementary information inherent in **multi-channel PSG signals**. For instance, the interplay between EEG (reflecting cortical activity) and EOG (capturing rapid eye movements) is crucial for accurately distinguishing sleep stages like REM, a nuance often lost in channel-agnostic models Martin et al. (1972).

To bridge this critical gap, we propose the first Multi Channel, Multi Source Unsupervised Domain Adaptation framework specifically designed for robust sleep staging. Our framework is engineered to simultaneously tackle the challenges of heterogeneous data sources and complex inter signal dependencies. At its core, we introduce a context aware encoder structure, enhanced by a novel Multi Scale Feature Collaboration Pauly et al. (2003) and Squeeze and Excitation (MSFC+SE) module Hu et al. (2018), to dynamically model inter channel relationships. By jointly optimizing for representation learning and domain alignment across multiple sources, our model learns features that are both discriminative and invariant to domain specific shifts. Extensive experiments on challenging cross dataset benchmarks demonstrate that our method achieves a state of the art accuracy of 77.49%, significantly outperforming existing baselines and validating its potential for robust clinical deployment. As summarized in Fig. 1, our multi source multi channel approach sits at the top of the historical landscape, outperforming non adaptation, single source, and multi source single channel methods.

The main contributions of our proposed MMUDA can be summarized as follows:

1. We pioneer a novel framework that, for the first time, addresses the combined challenges of multi-channel signal fusion and multi-source domain adaptation for automated sleep staging.

2. We design a new context-aware encoder with an MSFC+SE module to explicitly capture complementary information across physiological signals and effectively align feature representations from disparate domains.

3. We establish a new state-of-the-art on multiple public sleep datasets, demonstrating through extensive experimentation that our method offers superior generalization and robustness in realistic cross-domain scenarios.

## 2 METHOD

### 2.1 OVERVIEW

We propose a novel multi-channel, multi-source unsupervised domain adaptation framework (Fig. 2) designed to learn domain-invariant and class-discriminative representations from multiple labeled source domains and transfer the learned knowledge to an unlabeled target domain. The framework consists of four core components: the Multi-scale Temporal Feature Encoder (Sec. 2.2), Reconstructive Feature Learning (Sec. 2.3), Cross-domain Category Structure Learning (Sec. 2.4), and Multi-source Domain Alignment (Sec. 2.5). All components are jointly optimized in an end-to-end fashion. The overall training objective is summarized in Sec. 2.6. In the following derivative of MMUDA, let $\mathcal{D} = \{S_1, \ldots, S_N, T\}$ denote the domains with $D = |\mathcal{D}| = N+1$; unless otherwise stated, pairwise alignment terms such as $L_{\mathrm{mmd}}$ average over all unordered pairs $(i, j)$ with $X^{(i)}, X^{(j)} \in \mathcal{D}$ (including target $T$).

### 2.2 MULTI-SCALE TEMPORAL FEATURE ENCODER

The Multi scale Temporal Feature Encoder is designed to extract both localized and contextual temporal information from EEG signals. It integrates two key components: the Multi scale Feature Extraction CNN (MSFC) and the Temporal Context Encoder (TCE). The MSFC comprises two parallel

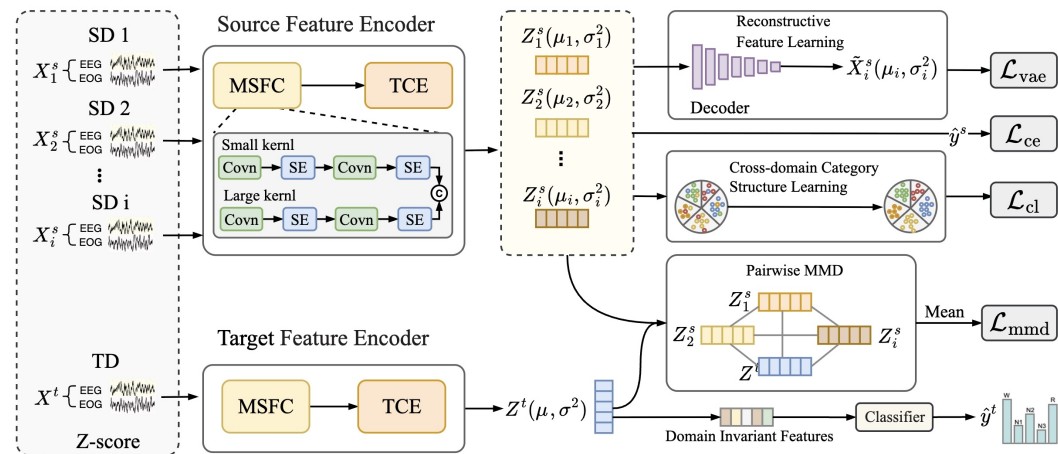

Figure 2: Overall framework of the proposed multi-source multi-channel unsupervised domain adaptation model for sleep staging.

convolutional pathways with distinct kernel sizes, enabling the capture of diverse temporal dependencies ranging from short term local patterns to long range signal dynamics. Each convolutional path is enhanced with a Squeeze and Excitation (SE) block, which applies modality aware channel wise attention to selectively emphasize discriminative features across both EEG and EOG channels. By adaptively recalibrating channel responses based on their global importance, the SE blocks facilitate effective fusion of complementary modalities. The detailed architecture and hyperparameters of the Multi scale Temporal Feature Encoder are summarized in Tab. 5. Following this, the TCE utilizes a bidirectional LSTM to encode temporal context from the concatenated multi scale features, effectively modeling sequential dependencies in both forward and backward directions. This hierarchical design facilitates the learning of robust, temporally aware feature representations essential for downstream classification and domain adaptation.

## 2.3 Reconstructive Feature Learning

To enable the model to capture comprehensive and informative latent representations from the source domains, we incorporate a Reconstructive Feature Learning Module based on the Variational Autoencoder framework. This module serves as an unsupervised regularizer that encourages the encoder to preserve essential input information while mapping it to a continuous latent space. By reconstructing the original input from the latent representation, the model is guided to retain semantic content and avoid collapsing to trivial or domain-specific features. Given an input sample $x$, an encoder network $\text{Enc}(\cdot)$ outputs the mean and log-variance of the latent distribution:

$$\mu, \log \sigma^2 = \text{Enc}(x) \tag{1}$$

The latent vector $z$ is sampled using the reparameterization trick:

$$z = \mu + \sigma \cdot \epsilon, \quad \epsilon \sim \mathcal{N}(0, I) \tag{2}$$

The sampled latent representation $z$ is then fed into two branches: (1) a decoder $\text{Dec}(z)$ for reconstructing the input, and (2) a classifier for source domain label prediction.

The VAE loss combines reconstruction loss and Kullback-Leibler divergence with diagonal Gaussian posterior:

$$\mathcal{L}_{\text{vae}} = \|x - \hat{x}\|_2^2 + \text{KL}\left(\mathcal{N}(\mu, \text{diag}(\sigma^2)) \| \mathcal{N}(0, I)\right) \tag{3}$$

This module enforces a continuous latent space, which is beneficial for both representation learning and domain alignment.

## 2.4 Cross-domain Category Structure Learning

Different sleep stages exhibit distinct physiological patterns, which should be consistently captured across domains to enable effective knowledge transfer. However, due to domain shifts caused by differences in recording conditions, devices, or populations, the same class (e.g., N2) might appear differently in different domains. To address this challenge, we propose a Cross-domain Category Structure Learning module that aims to preserve class-wise semantic consistency across domains by encouraging domain-invariant and discriminative feature representations.

Specifically, we adopt a supervised contrastive learning strategy that brings together samples of the same class while pushing apart those from different classes. Given a mini-batch of normalized feature vectors, we compute their pairwise cosine similarity matrix and apply a temperature-scaled softmax to obtain the contrastive logits. Positive pairs are defined as instances sharing the same class label. In addition, to enhance cross-domain generalization, we optionally restrict positive pairs to come from different domains only, thereby focusing on aligning the intra-class structure across domains.

---

**Algorithm 1** MMUDA: Multi-channel Multi-source Unsupervised Domain Adaptation

---

**Input:** Source domains $\{\mathcal{X}_S^{(i)}, \mathcal{Y}_S^{(i)}\}_{i=1}^N$, target domain $\mathcal{X}_T$

**Output:** Trained model $\mathcal{F}_\theta$

**Multi-scale Temporal Encoding for** $x \in \mathcal{X}_S \cup \mathcal{X}_T$ **do**

> Extract MSFC features  Fuse modalities via SE blocks  Encode context via Bi-LSTM

**Reconstructive Feature Learning for** $x \in \mathcal{X}_S$ **do**

> Encode: $\mu, \log \sigma^2 = \text{Enc}(x)$  Sample: $z = \mu + \sigma \cdot \epsilon, \epsilon \sim \mathcal{N}(0, I)$  Reconstruct: $\hat{x} = \text{Dec}(z)$  Compute $\mathcal{L}_{\text{vae}}$ (Eq. 3)

**Category Structure Learning foreach** *mini-batch from* $\mathcal{X}_S$ **do**

> Compute embeddings $\mathbf{z}_i$, labels $y_i$  Identify positives across domains  Compute $\mathcal{L}_{\text{cl}}$ (Eq. 4)

**Multi-source Domain Alignment foreach** *domain pair* $(i,j)$ **do**

> Compute $\text{MMD}^2(\mathcal{X}^{(i)}, \mathcal{X}^{(j)})$

Aggregate: $\mathcal{L}_{\text{mmd}}$ (Eq. 6)

**Optimization** Compute $\mathcal{L}_{\text{ce}}$, total loss $\mathcal{L}_{\text{total}}$ (Eq. 7)  Update parameters $\theta$

**return** $\mathcal{F}_\theta$

---

Formally, given a batch of feature embeddings $\{\mathbf{z}_i\}_{i=1}^N$, their corresponding labels $\{y_i\}_{i=1}^N$, and optional domain IDs $\{d_i\}_{i=1}^N$, the loss is computed as:

$$\mathcal{L}_{\text{cl}} = -\frac{1}{|\mathcal{V}|} \sum_{i \in \mathcal{V}} \frac{1}{|\mathcal{P}(i)|} \sum_{j \in \mathcal{P}(i)} \log \frac{\exp(\mathbf{z}_i^\top \mathbf{z}_j / \tau)}{\sum_{k \in \mathcal{A}(i)} \exp(\mathbf{z}_i^\top \mathbf{z}_k / \tau)}, \quad (4)$$

where $\tau$ is the temperature parameter, $\mathcal{P}(i)$ denotes the set of positive indices for anchor $i$, $\mathcal{A}(i)$ is the set of all indices excluding $i$, and $\mathcal{V}$ is the set of anchors with at least one positive. When the cross_domain_only option is enabled, $\mathcal{P}(i)$ is restricted to samples of the same class from a different domain than $i$. We use the $\ell_2$-normalized embedding $z = f_\theta(x)/\|f_\theta(x)\|_2$ in the contrastive loss, and the sets $P(i)$, $A(i)$, and $V$ follow the construction described above.

## 2.5 Multi-source Domain Alignment

Addressing distributional discrepancies among multiple labeled source domains and the unlabeled target domain necessitates precise alignment of their underlying feature distributions. We propose a **Multi-source Domain Alignment** module that explicitly performs *pairwise distribution alignment* across all domains. Unlike methods that align source domains collectively or focus solely on source–target adaptation, our approach systematically computes the alignment loss between every domain pair, ensuring fine-grained and symmetric adaptation across the entire domain set.

The alignment mechanism leverages the MMD Gretton et al. (2012), a kernel-based, non-parametric statistical measure that captures distributional divergence in reproducing kernel Hilbert space (RKHS). Given two sets of features $\mathcal{X} = \{x_i\}_{i=1}^n$ and $\mathcal{Y} = \{y_j\}_{j=1}^m$ from two different domains, the empirical MMD is computed as:

$$\text{MMD}^2(\mathcal{X}, \mathcal{Y}) = \frac{1}{n^2} \sum_{i=1}^n \sum_{j=1}^n k(x_i, x_j) + \frac{1}{m^2} \sum_{i=1}^m \sum_{j=1}^m k(y_i, y_j) - \frac{2}{nm} \sum_{i=1}^n \sum_{j=1}^m k(x_i, y_j), \quad (5)$$

where we adopt a multi-kernel RBF, $k(x, y) = \sum_{q=1}^{Q} \exp\left(- \|x - y\|_2^2/(2\sigma_q^2)\right)$, with $\{\sigma_q\}_{q=1}^{Q}$ fixed on a log-spaced grid. For $D$ total domains, the overall loss is computed by averaging the MMD values over all $\binom{D}{2}$ domain pairs:

$$\mathcal{L}_{\text{mmd}} = \frac{2}{D(D-1)} \sum_{1 \leq i < j \leq D} \text{MMD}^2(\mathcal{X}^{(i)}, \mathcal{X}^{(j)}), \quad (\mathcal{X}^{(i)}, \mathcal{X}^{(j)}) \in \mathcal{D}. \tag{6}$$

Through this pairwise alignment strategy, the model is explicitly encouraged to reduce inter-domain variation, thereby promoting consistent and semantically meaningful feature representations that are robust to domain shifts.

## 2.6 Overall Objective

As summarized in Section 2.4, the training process integrates multiple components, including multi-scale temporal encoding, reconstructive feature learning, cross-domain category structure learning, and multi-source domain alignment. The overall loss function is a weighted combination of these objectives:

$$\mathcal{L}_{\text{total}} = \mathcal{L}_{\text{ce}} + \lambda_{\text{vae}}\mathcal{L}_{\text{vae}} + \lambda_{\text{cl}}\mathcal{L}_{\text{cl}} + \lambda_{\text{mmd}}\mathcal{L}_{\text{mmd}} \tag{7}$$

Here, $\mathcal{L}_{\text{ce}}$ denotes the cross-entropy loss computed on the labeled source domain, and $\lambda_{\text{vae}}, \lambda_{\text{cl}}, \lambda_{\text{mmd}}$ are hyperparameters that balance the contribution of each module.

## 3 EXPERIMENTS

### 3.1 Dataset and Data Processing

We used five public datasets to evaluate the proposed MMUDA model. The SleepEDF dataset Kemp et al. (2000) contains 100 PSG recordings from healthy subjects, with EEG sampled at 100 Hz; we used the Fpz Cz EEG and horizontal EOG channels in our experiments. The DCSM dataset Perslev et al. (2021) consists of 103 PSG recordings sampled at 256 Hz, where the F4-M1 EEG and E1-M2 EOG channels were used for evaluation. The ISRUC dataset Khalighi et al. (2016) provides 100 annotated PSG recordings at 200 Hz; we employed the C4-M1 EEG and E1-M2 EOG channels, using the annotations from expert 1. The HMC dataset Alvarez-Estevez & Rijsman (2021) includes 100 PSG recordings sampled at 256 Hz, and we used the F4-M1 EEG and E1-M2 EOG channels for our evaluation. Finally, the P2018 dataset Ghassemi et al. (2018) comprises 100 PSG recordings with a 200 Hz sampling rate, where we selected the C3-M2 EEG and E1-M2 EOG channels for model training and testing. A detailed summary of the datasets, including subject counts, sampling rates, and recording channels, is presented in Tab. 1, while the distribution of sleep stages across these datasets is reported in Tab. 6. In this study, all EEG signals were uniformly resampled to 100 Hz. Prior to model training, the sleep recordings underwent band pass filtering between 0.3 Hz and 35 Hz to remove high frequency noise and baseline drift, while preserving the physiological frequency bands relevant for sleep staging. Subsequently, Z score normalization was applied on a per recording basis to reduce inter subject variability and improve model generalization. Finally, sleep stage labels were standardized according to the AASM guidelines to maintain consistency across datasets. The distribution of sleep stages across the datasets is further detailed in Appendix D.

| Dataset | Subjects | Sampling Rate | Total Samples | EEG Channel | EOG Channel |
|---------|----------|---------------|---------------|-------------|-------------|
| SleepEDF | 100 | 100 | 115,480 | Fpz–Cz | Horizontal |
| DCSM | 103 | 256 | 101,480 | F4–M1 | E1–M2 |
| ISRUC | 100 | 200 | 89,240 | C4–M1 | E1–M2 |
| HMC | 100 | 256 | 90,280 | F4–M1 | E1–M2 |
| P2018 | 100 | 200 | 88,880 | C3–M2 | E1–M2 |

Table 1: Overview of the processed datasets.

| Method | Metrics | SleepEDF | DCSM | ISRUC | HMC | P2018 | AVG |
|--------|---------|----------|------|-------|-----|-------|-----|
| | | | *Single-Source Domain* | | | | |
| ADAST | ACC | 51.94±2.10 | 62.00±1.47 | 63.77±2.55 | 59.73±0.83 | 59.90±1.62 | 59.47±1.77 |
| | MF1 | 45.25±1.33 | 55.84±2.67 | 56.92±0.73 | 53.70±2.96 | 50.27±2.49 | 52.39±1.81 |
| TUDAMatch | ACC | 42.08±1.74 | 55.10±0.98 | 54.29±2.26 | 58.83±2.37 | 51.81±1.62 | 52.42±1.64 |
| | MF1 | 38.84±1.18 | 42.07±1.41 | 46.88±2.30 | 52.72±2.73 | 38.62±1.80 | 43.83±1.90 |
| SDA | ACC | 59.73±2.13 | 64.33±1.44 | 65.54±2.89 | 67.08±1.96 | 62.25±2.54 | 63.79±2.19 |
| | MF1 | 55.47±1.97 | 56.30±2.11 | 60.12±2.74 | 62.27±1.66 | 54.15±3.02 | 57.66±2.30 |
| | | | *Multi-Source Domain* | | | | |
| DANN | ACC | 74.61±1.25 | **81.83±2.01** | 76.95±1.46 | 74.29±2.61 | 73.47±2.88 | 76.23±2.05 |
| | MF1 | 70.55±2.04 | **77.19±0.96** | 73.18±2.68 | 70.36±1.41 | 69.77±1.87 | 72.21±1.72 |
| DSA | ACC | 72.57±2.84 | 77.25±1.59 | 75.01±2.47 | 71.82±3.73 | 70.55±1.27 | 73.44±2.52 |
| | MF1 | 67.57±0.85 | 72.40±1.69 | 68.82±2.91 | 67.62±1.88 | 66.04±2.49 | 68.49±2.70 |
| ADADSA | ACC | 74.90±1.45 | 79.46±3.63 | 75.67±0.95 | 71.42±2.57 | 71.48±1.11 | 74.59±2.31 |
| | MF1 | 70.15±1.25 | 74.66±2.74 | 71.44±2.01 | 68.04±0.92 | 67.56±2.66 | 70.37±1.96 |
| SleepDG | ACC | 76.94±1.17 | 81.08±1.71 | 76.74±1.99 | 74.88±1.34 | 72.41±1.83 | 76.41±1.69 |
| | MF1 | **72.45±2.50** | 74.97±3.07 | 73.16±0.89 | 71.57±2.75 | 69.58±1.53 | 72.34±2.11 |
| MUDA | ACC | 76.76±1.79 | 76.57±2.81 | 75.77±1.10 | **75.37±2.46** | 78.07±0.85 | 76.51±1.97 |
| | MF1 | 69.25±2.99 | 71.28±1.58 | 72.26±2.14 | 71.28±1.44 | 72.55±2.39 | 71.32±2.22 |
| **Ours** | **ACC** | **77.62±2.11** | 81.68±1.80 | **77.58±1.47** | 75.33±1.98 | **75.24±2.27** | **77.49±1.94** |
| | **MF1** | 72.28±2.70 | 76.96±1.09 | **73.90±2.56** | 71.82±1.33 | **72.35±1.77** | **73.46±2.01** |

Table 2: Comparison with DA methods across various source and target domain combinations. AVG ± values are calculated as the RMS of the respective domain-specific standard deviations.

## 3.2 EXPERIMENTAL RESULTS

### 3.2.1 COMPARISON WITH DA METHODS

We compare our method against a range of state-of-the-art domain adaptation approaches, including both single-source and multi-source domain adaptation baselines. ADAST and TUDAMatch are single-source methods, where only one labeled source domain is used for adaptation. ADAST Eldele et al. (2022) employs adversarial training to align feature distributions between the source and target domains, while TUDAMatch Luo et al. (2023) incorporates uncertainty-aware pseudo-labeling to enhance adaptation to the target domain. For a fair comparison, we individually adapt each source domain to the target and report the average performance. In contrast, DANN Ganin et al. (2016), DSA Fan et al. (2022), and ADADSA Fan et al. (2022) are multi-source approaches that utilize labeled data from multiple heterogeneous source domains to improve generalization to unseen targets. Specifically, DANN applies adversarial learning via a domain discriminator to extract domain-invariant representations; DSA aligns attention-enhanced class prototypes across domains; and ADASDA uses statistical distribution matching based on MMD. To ensure a fair comparison with our method, we modified these baselines to support a flexible number of source domains.

The performance of these baselines and our proposed method is compared across five benchmark target domains: SleepEDF, DCSM, ISRUC, HMC, and P2018, and the results are comprehensively summarized in Tab. 2. Our method achieves the high- est ACC (77.49%), surpassing all baselines.Our method consistently achieves the best performance on all target domains in terms of both ACC and MF1 score, attaining the highest overall averages of 77.49% and 73.46%, respectively. Compared to the strongest baseline, SleepDG Wang et al. (2024), our approach improves average ACC by +1.08% and MF1 by +1.12%, while maintaining comparable or lower variance. Notably, these gains are particularly prominent on domains exhibiting substantial distribution shifts, such as P2018 and HMC, where our model surpasses all competitors with MF1 scores of 72.35% and 71.82%. This outstanding performance is attributed to our framework's ability to integrate supervision from diverse labeled source domains through a multi-source domain learning strategy, utilize a shared VAE-based encoder to model structured multi-channel physiological signals such

as EEG, enhance class-discriminative alignment under cross-domain constraints via a supervised contrastive learning objective, and reduce distributional discrepancies in the latent space through domain-invariant feature alignment based on MMD.

### 3.2.2 COMPARISON WITH NON-DA METHODS

| Method | Metrics | SleepEDF | DCSM | ISRUC | HMC | P2018 | AVG |
|---|---|---|---|---|---|---|---|
| DeepSleep | ACC | 76.53±2.50 | 71.69±3.40 | 71.61±2.22 | 63.06±2.82 | 53.45±1.26 | 67.27±2.54 |
| | MF1 | 60.65±4.80 | 60.70±3.83 | 63.89±3.10 | 50.69±4.14 | 46.34±4.80 | 56.45±4.18 |
| TinySleep | ACC | 76.09±3.93 | 74.31±1.08 | 68.84±2.73 | 64.61±1.80 | 51.65±4.86 | 67.10±3.19 |
| | MF1 | 60.12±3.39 | 63.35±4.88 | 61.58±2.16 | 51.61±3.06 | 45.69±4.23 | 56.47±3.67 |
| AttnSleep | ACC | 71.10±1.62 | 75.31±4.33 | 70.06±3.45 | 70.68±3.37 | 68.54±2.22 | 71.14±3.15 |
| | MF1 | 64.53±2.10 | 70.22±4.81 | 64.86±3.14 | 65.70±1.22 | 65.95±2.20 | 66.25±2.97 |
| SeriesSleep | ACC | 68.65±3.14 | 74.56±2.49 | 73.81±1.90 | 68.47±4.10 | 69.68±4.38 | 71.03±3.26 |
| | MF1 | 63.49±3.70 | 69.62±2.93 | 72.19±1.86 | 66.20±2.96 | 69.96±3.91 | 68.29±3.28 |
| **Ours** | **ACC** | **77.62±2.11** | **81.68±1.80** | **77.58±1.47** | **75.33±1.98** | **75.24±2.27** | **77.49±1.94** |
| | **MF1** | **72.28±2.70** | **76.96±1.09** | **73.90±2.56** | **71.82±1.33** | **72.35±1.77** | **73.46±2.01** |

Table 3: Comparison with non-DA methods across the same domain adaptation settings.

Tab. 3 reports the comparison between our proposed method and several state-of-the-art non-domain generalization (non-DA) baselines, including *DeepSleep* Supratak et al. (2017), *TinySleep* Supratak & Guo (2020), *AttnSleep* Eldele et al. (2021), and *SeriesSleep* Lee et al. (2023), under the same multi-source unsupervised domain adaptation settings across five benchmark datasets. Our model consistently outperforms all baselines across all target domains in both ACC and MF1 metrics. Specifically, our approach achieves the highest overall average ACC of **77.49%** and MF1 score of **73.46%**, significantly surpassing the closest competitor, SeriesSleep, which records **71.03%** in ACC and **68.29%** in MF1.

The performance gains are especially notable in challenging domains such as *P2018* and *HMC*, where domain shifts are more severe. For instance, on the *P2018* dataset, our method improves MF1 by **+2.39%** over SeriesSleep and by over **+25.0%** compared to DeepSleep and TinySleep. These results highlight the robustness and adaptability of our model to complex, unseen target domains.

## 4 ABLATION STUDY

| Method | Metric | SleepEDF | DCSM | ISRUC | HMC | P2018 | AVG |
|---|---|---|---|---|---|---|---|
| BASE | ACC | 76.14±1.37 | 80.94±1.60 | 76.30±1.30 | 73.57±1.46 | 72.68±1.07 | 75.93±1.37 |
| | MF1 | 71.09±1.95 | 76.44±1.71 | 73.05±1.52 | 70.53±1.79 | 69.69±1.95 | 72.16±1.79 |
| BASE+CL | ACC | 76.42±1.73 | 81.12±1.02 | 76.79±1.43 | 73.75±1.20 | 73.83±1.97 | 76.38±1.51 |
| | MF1 | 71.37±1.60 | 76.69±1.97 | 73.36±1.29 | 70.39±1.51 | 71.85±1.81 | 72.73±1.65 |
| BASE+MMD | ACC | 76.34±1.16 | 81.26±1.83 | 76.94±1.61 | 73.93±1.59 | 72.71±1.30 | 76.24±1.52 |
| | MF1 | 71.30±1.73 | 75.90±1.46 | 73.09±1.89 | 70.89±1.21 | 70.81±1.36 | 72.40±1.56 |
| BASE+VAE | ACC | 77.37±1.69 | 81.12±1.61 | 77.21±1.36 | 74.39±1.22 | 74.91±1.65 | 77.00±1.50 |
| | MF1 | 72.25±1.26 | 76.54±1.42 | 73.38±1.37 | 71.10±1.51 | 73.02±1.73 | 73.26±1.47 |
| **Ours** | **ACC** | **77.62±2.11** | **81.68±1.80** | **77.58±1.47** | **75.33±1.98** | **75.24±2.27** | **77.49±1.94** |
| | **MF1** | **72.28±2.70** | **76.96±1.09** | **73.90±2.56** | **71.82±1.33** | **72.35±1.77** | **73.46±2.01** |

Table 4: Ablation study results comparing the base model and its variants. AVG ± values are computed using root mean square of the domain-specific standard deviations.

Tab. 4 presents the results of our ablation study, aiming to disentangle the individual contributions of each component in our proposed framework. We compare the full model (**Ours**) with several variants: the base model (BASE), and versions augmented with contrastive learning (BASE+CL),

domain-invariant alignment via MMD (BASE+MMD), and VAE-based multi-channel modeling (BASE+VAE).

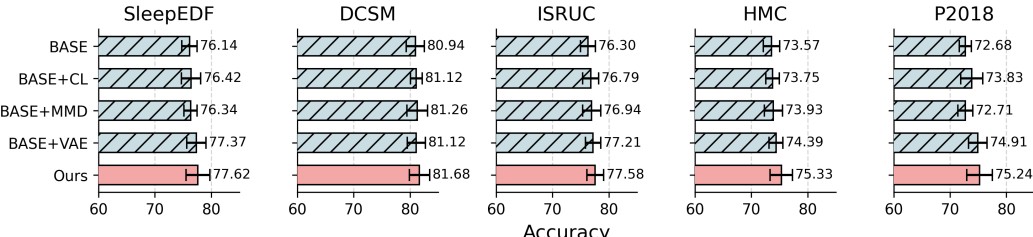

Figure 3: Ablation study results on five public datasets, comparing the base model (BASE) and its variants with our full model.

The base model already achieves solid performance, with an average ACC of 75.93% and MF1 of 72.16%. Adding supervised contrastive learning leads to a performance gain, particularly in MF1 (+0.57%), indicating improved class discriminative representation. Introducing MMD alone results in marginal changes, suggesting its primary benefit lies in enhancing distribution alignment rather than discriminative capability. Notably, incorporating the VAE based encoder (BASE+VAE) yields substantial improvements in both ACC (77.00%) and MF1 (73.26%), demonstrating the effectiveness of structured multi channel representation learning for physiological signals. Finally, the full model that combines all components achieves the highest performance, 77.49% in ACC and 73.46% in macro F1, outperforming all ablated variants, as illustrated in Fig. 3. These results confirm the complementary benefits of each module and the synergy among them in facilitating robust and generalized cross domain sleep staging. To further validate the learned representations, we provide feature visualizations across domains and sleep stages in Appendix E.

## 5 CASE STUDY

For a more comprehensive assessment of the practical effectiveness of our method, we conduct a case study using a real-world clinical dataset collected from form a collaborative uinversity hospital. The dataset comprises 45 overnight PSG recordings with corresponding clinical annotations, obtained from consenting patients between September 2022 and August 2024. These recordings are treated as the target domain (collaborative uinversity hospital), while five public datasets DCSM, ISRUC, HMC, P2018, and SleepEDF are used as the source domains. As shown in Fig. 4, our method significantly outperforms the baseline in both ACC and MF1. Specifically, our model achieves an ACC of 71.1% and a MF1 of 62.15%, compared to 63.07% and 53.79% for the baseline, respectively. These improvements demonstrate the effectiveness of our domain adaptation strategy in real clinical settings, where data distribution shifts are inevitable and labeled target samples are scarce.

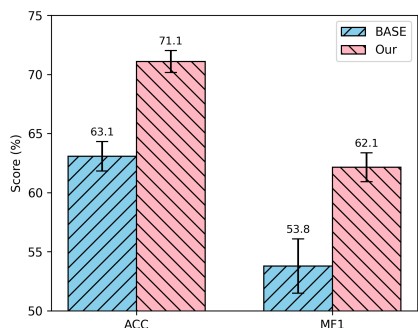

Figure 4: Performance comparison with the BASE model.

To further illustrate the effectiveness of our method, we visualize the learned feature representations of the adaptation task using the t-SNE method Maaten & Hinton (2008). Fig. 5(a) and (b) show the feature distribution of the baseline model and our proposed model from the perspective of domain alignment, where different colors represent different source and target domains. Compared with Fig. 5(a), where domain-specific clusters are still separated, Fig. 5(b) demonstrates that our method achieves better overlap between source and target domains, indicating improved domain-invariant representation learning. Fig. 5(c) and (d) present the same features but viewed from the class perspective, where different colors represent sleep stages (Wake, N1, N2, N3, and REM). As shown in Fig.(d), samples belonging to the same sleep stage form more compact and clearly separable clusters compared to Fig. 5(c). This trend demonstrates that our method simultaneously enhances domain alignment and class discriminability, which are critical for robust cross-domain sleep staging.

Fig. 6 presents a comparison between the manual scoring annotated by expert one (red) and the automated sleep stage classification generated by our proposed model (blue) for a representative subject from the EEH-85 dataset. The two hypnograms exhibit a high degree of overlap. Minor discrepancies primarily occur at transition boundaries (e.g., between N1 and N2 or REM and N2), where frequent stage shifts naturally increase classification difficulty. Despite these local inconsistencies, the overall sleep architecture produced by our model closely mirrors the expert labels, demonstrating both quantitative ACC and qualitative reliability in capturing sleep dynamics.

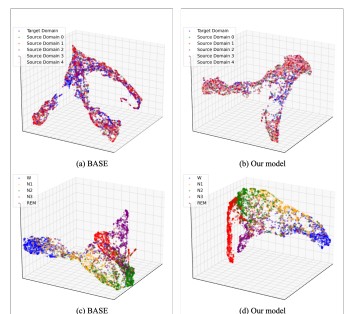

Figure 5: features visualization.

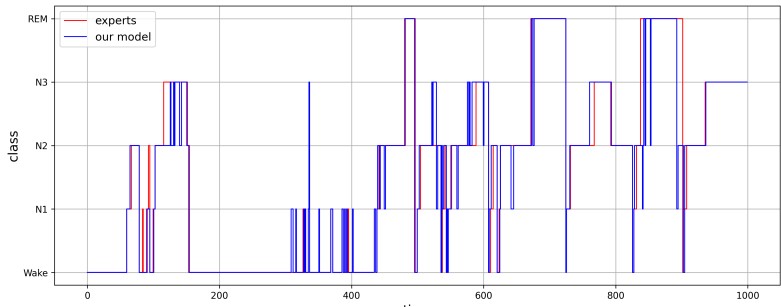

Figure 6: Comparison between the manual scoring annotated by clinical expert 85 (blue) and the automated scoring generated by the proposed model (red) for Subject 85 from the EEH dataset.

## 6 CONCLUSION

In this work, we analysis of five benchmark datasets revealed substantial discrepancies in class balance and stage composition, as well as broader inconsistencies in recording environments, subject demographics, sensor configurations, and signal quality. To overcome these challenges, we introduced MMUDA, a multi-source multi-channel unsupervised domain adaptation framework that leverages diverse datasets to learn domain-invariant and robust feature representations. Extensive experiments on five benchmark datasets and locally collected hospital data demonstrate that MMUDA effectively mitigates domain shif, substantially improves the recognition of underrepresented stages. Achieving state-of-the-art performance in cross-domain sleep staging without requiring target domain labels, our approach highlights the importance of multi-source adaptation for practical and clinically relevant deployment.

## 7 FUTURE WORK

We proposed the MMUDA framework, which leverages multi-source domains, temporal context modeling, and distribution alignment to alleviate domain shifts. In future work, we plan to extend MMUDA to a broader range of EEG-based applications (e.g., emotion recognition, fatigue detection, and neurological disorder diagnosis) and explore its potential for real-time adaptive deployment in clinical environments. Furthermore, incorporating multimodal signals such as EOG, EMG, or respiratory activity, designing personalized adaptation strategies to address intra-individual variability, and applying more advanced alignment techniques (e.g., adversarial learning or optimal transport) are promising directions. Finally, integrating semi-supervised or few-shot learning and improving model efficiency for wearable or mobile devices will further enhance the practical value of MMUDA.

ETHICS STATEMENT

This study makes use of publicly accessible sleep datasets (SleepEDF, DCSM, ISRUC, HMC, P2018) that are widely employed in the literature. The approach presented here is intended strictly for research and methodological exploration. It is not certified for direct clinical application, and any future medical deployment would necessitate thorough regulatory review and oversight by qualified professionals.

REPRODUCIBILITY STATEMENT

We provide detailed descriptions of the datasets, model design, training procedures, and evaluation settings in the Methods and Experiments sections. Information on hyperparameters, preprocessing pipelines, and model variants is reported explicitly. To facilitate replication and extension, we will release the accompanying codebase so that the community can reproduce our results and build upon them.

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

## A   RELATED WORKS

### A.1   SLEEP STAGE CLASSIFICATION

Deep learning (DL) has been widely applied to sleep data analysis, and researchers have proposed a variety of end-to-end approaches that can automatically learn features directly from raw or multi-modal physiological signals (e.g., EEG, EOG, ECG, and PPG) while capturing temporal dependencies. Existing methods can generally be categorized into two groups: the first relies primarily on

convolutional neural networks (CNNs) Chambon et al. (2018) to extract intra-epoch local features, followed by recurrent neural networks (RNNs, such as LSTMs) Phan et al. (2018) or self-attention mechanisms to model inter-epoch contextual dependencies; the second employs hierarchical RNNs, Transformers, or graph neural networks to extract sleep features and transition patterns from multi-view, multi-scale, or global contexts. Representative works include DeepSleepNet Supratak et al. (2017), SeqSleepNet Phan et al. (2019), SleepTransformer Eldele et al. (2021), SPSleep Zhang et al. (2025) and SleepFM Thapa et al. (2024), which leverage multi-branch convolution, multi-scale feature extraction, or attention mechanisms to effectively model salient local waveforms as well as cross-epoch sleep transition rules.

## A.2    Domain Adaptation for Sleep Stage Classification

Domain adaptation (DA) has been widely explored to address distribution shifts between source and target domains, and can be categorized as supervised, semi-supervised, or unsupervised (UDA) Pan & Yang (2009). Adversarial-based approaches aim to learn domain-invariant representations by confusing a domain discriminator, with extensions such as multi-class discriminators, attention-enhanced adversarial training, and multi-discriminator designs for global and subject-level alignment Gao et al. (2023). Alternatively, discrepancy-based methods align source and target features by minimizing statistical differences, e.g., MMD Gretton et al. (2012), CORAL Sun & Saenko (2016), Wasserstein distance, or similarity metrics. Recently, multi-source unsupervised domain adaptation (MUDA) has gained increasing attention. For example, SleepDG Wang et al. (2024) improves feature alignment by jointly minimizing first- and second-order statistical discrepancies. Another line of work, such as MUDAEEG Zhu et al. (2023), introduces domain-specific branches for each source target pair, combined with a domain-invariant branch and an adaptive weighting strategy, to achieve robust MUDA for EEG-based sleep staging. In contrast, our method leverages a larger set of heterogeneous source domains, including a real-world local clinical dataset, and enhances feature analysis by jointly exploiting EEG and EOG modalities for richer and more discriminative representations.

In sleep staging, researchers have also investigated personalized and cross-subject adaptation strategies. A notable example is BrainUICL Zhou et al. (2025), an EEG-based unsupervised continual learning framework, which employs Cross Epoch Alignment to dynamically adapt during long-term individual learning, thereby improving both adaptability and generalization.

## B    Feature Encoder Details

## C    Experimental Settings

Our proposed model is implemented using the `PyTorch` framework and trained with the Adam optimizer. The learning rate is set to $5 \times 10^{-3}$, and training is performed for 80 epochs with a batch size of 64. A dropout rate of 0.1 is applied to prevent overfitting. The input sequence length is set to $L = 20$, and the feature dimension is $d = 512$. We iteratively select four out of the five datasets as source domains, with the remaining one used as the target domain. ACC and MF1 are used as the primary evaluation metrics. All experiments are conducted on an NVIDIA RTX 3080 Ti GPU.

## D    Dataset Details

Tab 6 shows the distribution of sleep stages across six datasets. Clear differences can be observed: SleepEDF and DCSM are dominated by N2 but differ in wake proportion; ISRUC and HMC contain substantially more deep sleep (N3); P2018 resembles DCSM yet with fewer N3 epochs; EEH is the most imbalanced, with over half of its samples in N2 and very limited N1 and N3. The pronounced inconsistencies in class distributions across datasets underscore the inherent challenges of cross dataset generalization. Models trained on a single dataset are prone to overfitting towards its majority classes, while simultaneously exhibiting degraded performance on minority stages when transferred to datasets with markedly different distributions.

| Layer | Operation | Parameters |
|---|---|---|
| Small kernl | Conv2D-1 | (64, 50, 6) |
| | MaxPool2D-1 | (8, 1) |
| | SEBlock | channels=64, reduction=16 |
| | Conv2D-2 | (128, 8, 1) |
| | SEBlock | channels=128, reduction=16 |
| | Conv2D-3 | (128, 8, 1) |
| | MaxPool2D-2 | (4, 1) |
| Large kernl | Conv2D-1 | (64, 400, 50) |
| | MaxPool2D-1 | (4, 1) |
| | SEBlock | channels=64, reduction=16 |
| | Conv2D-2 | (128, 6, 1) |
| | SEBlock | channels=128, reduction=16 |
| | Conv2D-3 | (128, 6, 1) |
| | MaxPool2D-2 | (2, 1) |
| LSTM | Hidden Size | 256, Layers=1 |

Table 5: Hyper-parameters of the proposed Multi-Scale Temporal Feature Encoder. For Conv2D, the parameters from left to right are: (filters, kernel_size, stride).

| Dataset | W | N1 | N2 | N3 | REM | Total |
|---|---|---|---|---|---|---|
| SleepEDF | 31475 | 11129 | 45016 | 10375 | 17485 | 115480 |
| | 27.26% | 9.64% | 38.98% | 8.98% | 15.14% | 100% |
| DCSM | 20914 | 10191 | 43231 | 10322 | 16822 | 101480 |
| | 20.61% | 10.04% | 42.60% | 10.17% | 16.58% | 100% |
| ISRUC | 20684 | 11342 | 28029 | 17419 | 11766 | 89240 |
| | 23.18% | 12.71% | 31.41% | 19.52% | 13.18% | 100% |
| HMC | 15319 | 10381 | 33199 | 16937 | 14444 | 90280 |
| | 16.97% | 11.50% | 36.77% | 18.76% | 16.00% | 100% |
| P2018 | 15663 | 11395 | 37894 | 11881 | 12047 | 88880 |
| | 17.62% | 12.82% | 42.64% | 13.37% | 13.55% | 100% |
| EEH | 8301 | 3556 | 27578 | 3186 | 8999 | 51620 |
| | 16.08% | 6.89% | 53.43% | 6.17% | 17.43% | 100% |

Table 6: Sleep stage distribution across datasets, reported in both counts and percentages.

## E    FEATURES VISUALIZATION

To better understand the effectiveness of our proposed MMUDA framework, we perform a qualitative analysis of the learned feature representations using t-SNE. The visualizations illustrate how features from different datasets and sleep stages are distributed in the latent space. As shown in Fig. 7, we visualize the alignment between the target domain and multiple source domains across five datasets. In the baseline model, source and target features often remain misaligned, resulting in scattered and domain-specific clusters. In contrast, our method produces well-mixed feature distributions, where target-domain samples are more tightly aligned with their corresponding source-domain counterparts. This observation demonstrates that our multi-source alignment strategy effectively mitigates domain shift and encourages domain-invariant representations. Fig. 8, which illustrates the clustering of features according to sleep stages (Wake, N1, N2, N3, and REM). Compared to the baseline, our method yields more compact and clearly separated clusters, particularly for the minority stages (e.g., N1 and N3), which are typically harder to classify. This indicates that the model not only achieves cross-domain alignment but also preserves class-discriminative information, thereby improving stage-level separability.

## F    LLM USAGE

Large Language Models were used to assist in refining the manuscript's language. Their role was limited to tasks such as sentence rephrasing, grammar correction, and improving clarity and readability across sections, helping to enhance the overall flow of the text. The LLMs were not involved in developing ideas, research methodology, experimental design, or data analysis. All scientific content and findings were created by the authors, who take full responsibility for the manuscript. The use of LLMs complied with ethical standards, ensuring no plagiarism, misrepresentation, or scientific misconduct.

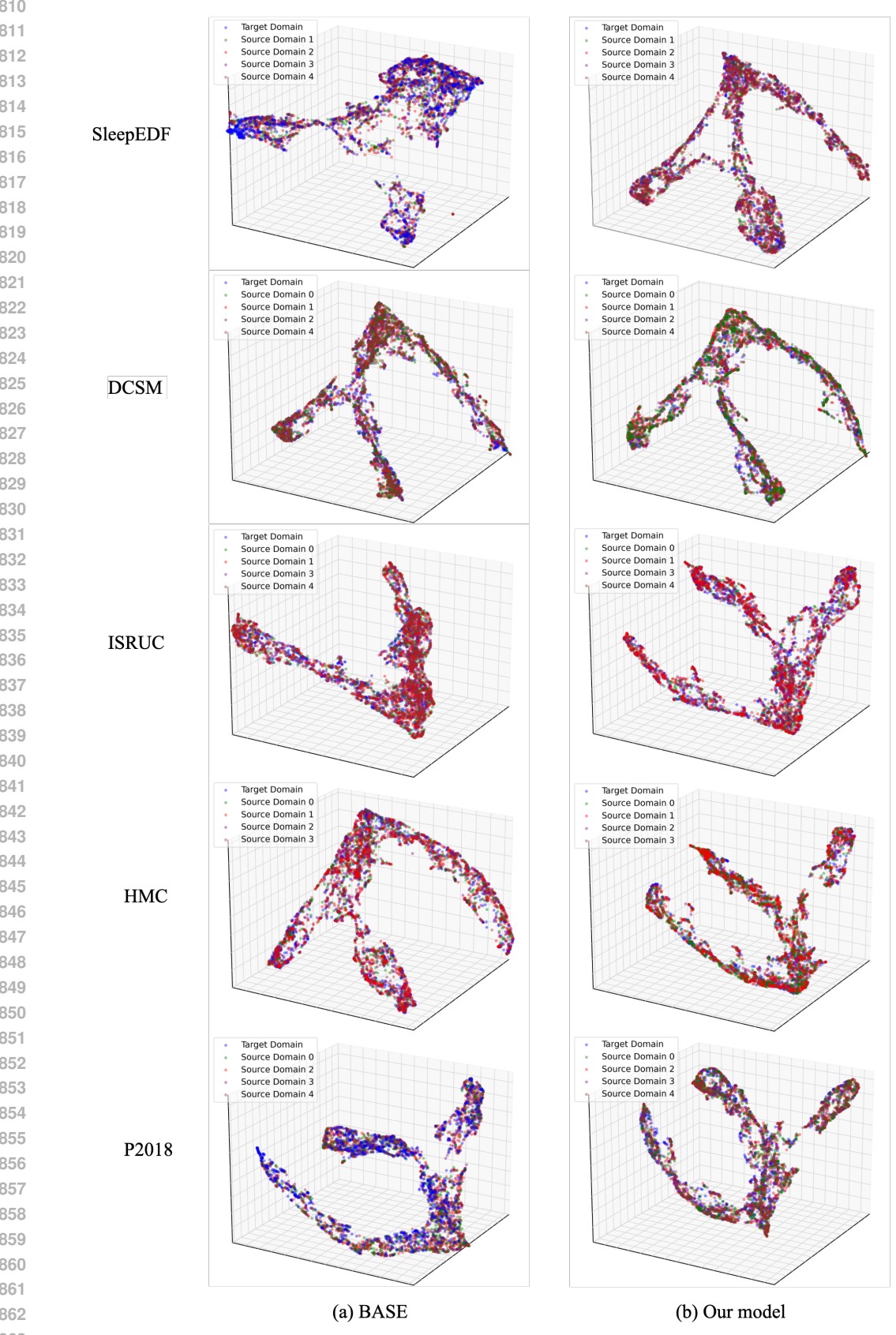

SleepEDF

DCSM

ISRUC

HMC

P2018

(a) BASE       (b) Our model

Figure 7: Visualization of Sleep features in different domains on five datasets.

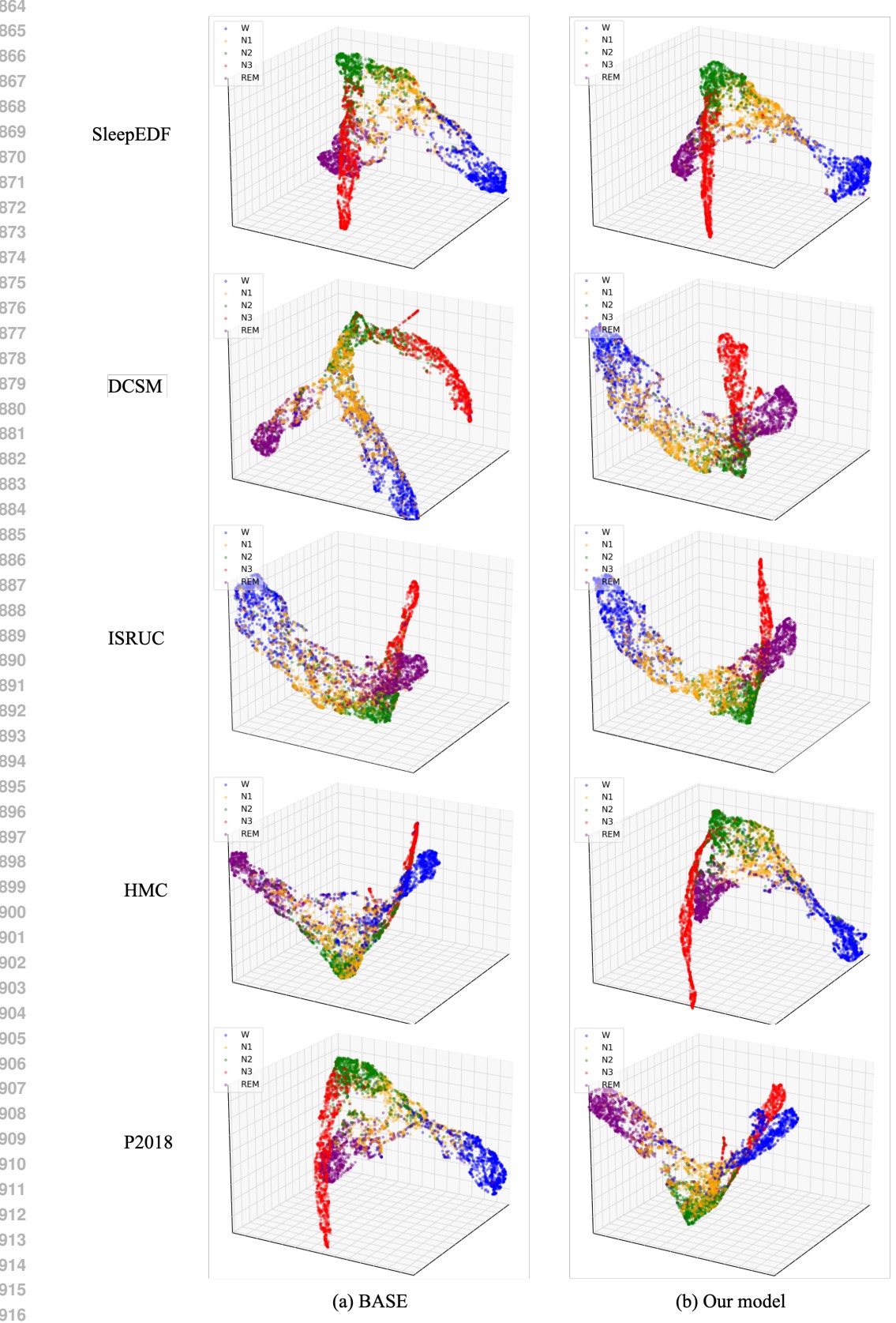

SleepEDF

DCSM

ISRUC

HMC

P2018

(a) BASE                                             (b) Our model

Figure 8: Visualization of Sleep Features for Different Sleep Stages on Five datasets.

