# OpenReview forum: "MMUDA: Towards Robust Sleep Staging with Multi-Source Multi-Channel Unsupervised Domain Adaptation"
_ICLR.cc/2026/Conference — Submitted to ICLR 2026_

### Official Review · Reviewer_bFZ2 · 2025-10-15

**Soundness:** 2
**Presentation:** 2
**Contribution:** 2
**Rating:** 2
**Confidence:** 4

**Summary:**

This paper proposes MMUDA, a multi-source and multi-channel unsupervised domain adaptation framework for sleep staging under clinical settings. The method employs a multi-scale temporal encoder with channel attention to capture both intra-channel temporal patterns and inter-channel dependencies in EEG/EOG signals. To align distributions across domains, it integrates a dual-alignment strategy—combining MMD-based explicit distribution alignment with supervised cross-domain contrastive learning to preserve class separability—while introducing a VAE constraint to ensure a compact latent space. The authors evaluate the framework across five public sleep datasets, performing extensive cross-domain experiments.

**Strengths:**

1.	Comprehensive cross-channel modeling: The combination of MSFC + SE with Bi-LSTM effectively captures local temporal patterns, contextual dependencies, and complementary information across modalities.
2.	Dual alignment design: The combination of MMD and contrastive learning provides a more complete and balanced approach to domain alignment.
3.	Extensive experimentation: The study covers five public datasets and an additional hospital dataset, demonstrating consistent results across diverse sources.

**Weaknesses:**

1.	Limited novelty: The proposed approach largely integrates existing ideas—MMD, contrastive learning, and VAE—which are already widely applied in sleep staging and domain adaptation research. The contribution is more in engineering integration than in conceptual innovation.
2.	Unrealistic problem setting: The assumption of having access to unlabeled target-domain data (typical in UDA) is idealized. In real clinical deployment, the target domain is often completely unseen, making domain generalization (DG) a more realistic and challenging scenario.
3.	Lack of theoretical analysis: The paper lacks theoretical justification or convergence analysis for its dual-alignment strategy or VAE-based regularization.
4.	Coarse cross-channel alignment: During alignment, the model operates on fused channel representations rather than aligning each modality or channel individually, which limits its ability to capture finer inter-modal discrepancies.
5.	Insufficient evaluation under missing labels: The assessment relies solely on overall ACC and MF1 metrics, without detailed error analysis of clinically critical boundaries (e.g., N1/N2, REM transitions) or uncertainty estimation, which are key for clinical reliability.
6.	Incomplete discussion of clinical deployability: While a hospital case study is included, the paper does not report inference efficiency, latency, or computational cost—factors crucial for evaluating real-world feasibility.

**Questions:**

Please see Weaknesses

**Details Of Ethics Concerns:**

NAN

---

### Official Review · Reviewer_yx5Z · 2025-10-17

**Soundness:** 2
**Presentation:** 2
**Contribution:** 1
**Rating:** 2
**Confidence:** 5

**Summary:**

The paper proposes MMUDA, a method for multi-source multi-channel unsupervised domain adaptation in sleep staging. It uses separate encoders with channel attention to model temporal and inter-channel information, and aligns source and target domains via Maximum Mean Discrepancy and cross-domain contrastive learning, stabilized by a variational autoencoder. Experiments on public datasets and in-house hospital data show improved cross-domain performance without target labels.

**Strengths:**

The method demonstrates modest performance gains over baseline approaches across five sleep staging datasets.

**Weaknesses:**

1. **A major concern is the apparent conflation between multi-source domain generalization (DG) and multi-source unsupervised domain adaptation (UDA).** The paper positions itself as a UDA method, yet several components—such as the VAE loss (Lvae​ ) and contrastive loss (Lcl​ )—are applied without using target-domain data, which aligns more closely with DG objectives. Only the pairwise MMD loss explicitly involves the target domain during training, creating a mismatch between the stated problem setting and the actual method design.

2. The choice of baselines is problematic. **In particular, SleepDG is a domain generalization method that assumes no access to target data at training time, making it an inappropriate comparator for a UDA approach like MMUDA**. This leads to an unfair comparison. Moreover, the reported performance gain over SleepDG is marginal—even though MMUDA leverages unlabeled target data during training, which should provide a significant advantage—further undermining the claimed effectiveness of the proposed method.

3. Section 3.2.2 lacks clear justification. Demonstrating that MMUDA outperforms non-DA sleep staging models does not validate the core contribution of the domain adaptation design, especially since the compared baselines are outdated. Recent strong sleep staging model such as BSTT[ICLR2023], CareSleepNet[JBHI2024], SleepSMC[ICLR2025] are missing from the evaluation.

4. Similarly, Figure 4 provides limited insight. Comparing MMUDA against an ablated “BASE” model on a new dataset does not convincingly demonstrate the value of the proposed adaptation components, as the baseline itself is not well-defined or contextualized within the current literature.

5. The ablation study is underdeveloped. Table 4 and Figure 3 present identical results, making one of them redundant. More importantly, the current ablations are framed relative to a poorly specified “BASE” model rather than the full MMUDA pipeline. A more informative analysis would systematically remove individual components—e.g., MMUDA, w/o CL, w/o VAE, w/o MMD—to isolate the contribution of each proposed module.

6. The method lacks clear technical novelty. For instance, the so-called "new context-aware encoder" is essentially a dual-branch convolutional structure combined with channel attention—a design pattern that has been widely adopted in prior work and does not constitute a meaningful architectural contribution.

7.  Minor presentation issues: In Figure 2, the label “Zt” is partially obscured by the diagram; in Figure 5, the font size is too small to be legible.

**Questions:**

In Algorithm 1, is adaptation to the target domain performed jointly with source-domain training? For domain adaptation, a more conventional and arguably more interpretable approach would be to first train a source model using only multi-source data, and then apply a separate adaptation stage to align it with the target domain. The current joint optimization setup raises questions about the actual role and necessity of each component in the adaptation process.

---

### Official Review · Reviewer_Euk2 · 2025-10-20

**Soundness:** 2
**Presentation:** 2
**Contribution:** 1
**Rating:** 2
**Confidence:** 5

**Summary:**

This paper proposes a framework called MMUDA to address domain shift in automated sleep staging across heterogeneous datasets and recording setups. MMUDA learns domain-invariant and class-discriminative representations by integrating multi-scale temporal encoding, variational autoencoder-based feature reconstruction, cross-domain contrastive learning, and Maximum Mean Discrepancy alignment across multiple labeled sources and an unlabeled target domain. Experiments on five public datasets demonstrate that MMUDA achieves better performance, outperforming both single- and multi-source baselines in cross-domain generalization.

**Strengths:**

- This paper target the real clinical challenge: domain shift across hospitals, devices, and populations limits the real-world reliability of sleep-staging AI systems.

**Weaknesses:**

- The technical novelty is incremental. The proposed MMUDA framework mainly combines existing techniques—MMD-based domain alignment, contrastive learning, and VAE regularization—in a straightforward manner. The methodological contribution lacks substantial theoretical innovation or principled integration of these components.

- Multi-source domain adaptation is a well-established and extensively studied area in machine learning [1,2,3]. However, in this paper, it remains unclear what makes MSDA for sleep staging uniquely challenging compared to other applications. For instance, do physiological signals such as EEG and EOG introduce distinct types of domain shifts (e.g., sensor-specific noise, inter-subject variability, temporal drift, or annotation inconsistency) that conventional MSDA methods fail to handle? Without a clear articulation of these domain-specific difficulties, it is difficult to assess the necessity and originality of MMUDA beyond being an adaptation of standard MSDA techniques to a new dataset.

- The motivation focuses primarily on empirical domain shift problems (different hospitals, devices, or populations) but lacks a deeper theoretical discussion of why existing UDA frameworks fail in multi-source or multi-channel settings. It doesn’t explicitly analyze how domain discrepancy manifests in EEG/EOG signals (e.g., spectral, temporal, or patient-specific variability), missing an opportunity to ground the problem in physiological signal theory.

- This work omits several recent advances in domain adaptation. The most recent domain adaptation baseline is from 2023. This limits the positioning of MMUDA within the broader landscape of robust time-series modeling. Moreover, the authors claim to be the first to propose a multi-source unsupervised domain adaptation or generalization framework for sleep staging; however, this claim is not accurate [4,5].

- The paper claims reproducibility but does not release the implementation or code. Without public resources or detailed hyperparameter settings, independent verification and comparison are difficult.

- The method does not explicitly address key characteristics of physiological data distribution shift such as temporal drift, evolving cross-signal dependencies, or non-stationary distributions.

- The experimental evaluation omits comparison with recent strong domain adaptation/generalization baselines in both general machine learning and sleep staging research. Consequently, it is unclear whether MMUDA truly advances the state of the art beyond these more modern frameworks.

- There’s no detailed analysis of which sleep stages (e.g., N1, REM) remain challenging or misclassified, despite known class imbalance. A confusion matrix or per-stage F1 breakdown would provide insight into model bias and clinical reliability, but is missing.

References

[1] Mansour, Y., Mohri, M., & Rostamizadeh, A. (2008). Domain adaptation with multiple sources. Advances in neural information processing systems, 21.

[2] Peng, Xingchao, et al. "Moment matching for multi-source domain adaptation." Proceedings of the IEEE/CVF international conference on computer vision. 2019.

[3] Zhao, Han, et al. "Adversarial multiple source domain adaptation." Advances in neural information processing systems 31 (2018).

[4] Zhu, Yangxin, et al. "Multi-source unsupervised domain-adaptation for automatic sleep staging." 2023 IEEE International Conference on Bioinformatics and Biomedicine (BIBM). IEEE, 2023.

[5] Lee, Seungyeon, et al. "Domain Invariant Representation Learning and Sleep Dynamics Modeling for Automatic Sleep Staging." ACM Transactions on Computing for Healthcare (2023).

**Questions:**

- Could the authors clarify how MMUDA integrates MMD, contrastive learning, and VAE objectives in a novel or synergistic way, rather than as a sequential combination of standard techniques?

- Are there any theoretical insights (e.g., generalization bounds, complementary objectives, or empirical trade-offs) that justify this particular design choice over other integration strategies (e.g., adversarial or optimal-transport alignment)?

- How does MMUDA explicitly address temporal non-stationarity in EEG/EOG signals, such as gradual distribution drift or evolving inter-signal dependencies? Have the authors evaluated performance over time (e.g., across different sleep cycles or subjects) to demonstrate temporal robustness? Would incorporating temporal alignment mechanisms (e.g., temporal contrastive learning, recurrent adaptation, or test-time adaptation) improve generalization?

- Since multiple source domains are used, does each contribute equally to adaptation? Have the authors examined whether some domains hurt generalization (negative transfer)? Could a learned source-weighting mechanism (e.g., importance weighting or domain relevance estimation) further improve performance?

---

### Official Review · Reviewer_HBSE · 2025-10-23

**Soundness:** 2
**Presentation:** 2
**Contribution:** 2
**Rating:** 4
**Confidence:** 4

**Summary:**

Sleep staging is fundamental for diagnosing and understanding sleep disorders. However, applying deep learning to sleep staging faces a major challenge—**domain shift**, which leads to poor generalization across datasets and recording conditions. **Unsupervised Domain Adaptation (UDA)** has emerged as a promising approach to mitigate this issue. Nonetheless, existing UDA methods for sleep staging encounter two critical limitations: (1) difficulty in handling multiple heterogeneous data sources, and (2) challenges in effectively modeling multi-channel polysomnography (PSG) signals. To address these limitations, the authors propose a **Multi-Channel, Multi-Source Unsupervised Domain Adaptation (MMUDA)** framework. The core component of the proposed framework is a **context-aware encoder** that incorporates a **Squeeze-and-Excitation (SE)** module to enhance feature representation. By training across multiple data sources, the framework is designed to learn discriminative and domain-invariant features, thereby improving generalization performance in sleep staging tasks.

**Strengths:**

1.The topic of sleep staging is of significant practical importance, as advancements in this area can greatly facilitate the real-world implementation of automated sleep analysis technologies.

2.Cross-domain feature alignment represents a major challenge across numerous research domains. The alignment techniques explored in this work may provide valuable insights and inspiration for addressing similar challenges in other fields.

3.The authors have also introduced a new dataset, which is commendable. Even if public release is not currently possible, a more detailed description of its characteristics would still contribute meaningfully to the progress of this research area.

**Weaknesses:**

1.In the *Introduction*, the authors highlight three main contributions, emphasizing that MMUDA can integrate information across multi-channel signals and perform multi-source domain adaptation. However, the *Methods* section is not structured according to this logic, resulting in a lack of coherence between the stated contributions and the methodological presentation.

2.The *Overview* section does not provide a true conceptual overview; rather, it simply lists four components (Sections 2.2–2.5) without explaining why these components are necessary or how they are conceptually related.

3.The writing style of the *Methods* section primarily describes *what* was done to achieve certain goals, but fails to explain *why* each methodological decision was made. This absence of rationale weakens the scientific justification of the work.

**These three issues collectively form the main reasons for my *weak reject* recommendation.**

4.In Section 2.4, the authors state that the same sleep stage may exhibit different characteristics across domains, using N2 as an example. However, the example is insufficiently detailed. Expanding this discussion could better demonstrate the necessity of multi-domain feature alignment.

5.The authors also note that Z-score normalization improves generalization. However, it is unclear whether the improvement in multi-source domain adaptation originates from Z-score normalization or from MMUDA itself. This distinction is important and should be verified experimentally.

6.Additionally, the claimed contribution of capturing information from multi-channel signals lacks direct experimental evidence. While performance improvements over state-of-the-art methods are reported, such results may arise from various factors, not solely from enhanced multi-channel representation learning.

**Questions:**

1.In Section 2.4, the authors mention that the same category exhibits different characteristics across domains and use N2 as an example. Could the authors elaborate further on this point to better justify the necessity of multi-domain feature alignment?

2.Can the authors provide experiments verifying whether the benefits of multi-source domain adaptation stem from Z-score normalization or from MMUDA itself?

3.Which experimental results directly demonstrate that MMUDA effectively captures information from multi-channel signals?

4.The case study with feature visualization suggests that the proposed model achieves multi-domain feature alignment, but this may represent an isolated example. Could the authors include statistical evidence or quantitative analysis to substantiate that cross-domain alignment is achieved more generally?

5.In Figure 4, does the “base model” refer to the same model used in the ablation experiments? If so, why is it used as a comparison baseline? If not, the terminology should be clarified to avoid confusion.

---

### Meta-Review · Area_Chair_aRA7 · 2025-12-25

**Summary:**

The paper proposes MMUDA, a framework for multi-source multi-channel unsupervised domain adaptation (UDA) in sleep staging, integrating Maximum Mean Discrepancy (MMD), contrastive learning, and VAEs. While the reviewers acknowledge the clinical importance of the problem, they unanimously recommend rejection (Scores: 4, 2, 2, 2). The consensus is that the submission suffers from limited technical novelty (viewed as an incremental combination of existing components), flawed experimental design (missing recent SOTA baselines from 2023-2025 and inappropriate comparisons between UDA and DG settings), and methodological ambiguities (lack of theoretical justification and conflation of UDA/DG objectives).

**Reviewer Concerns:**

Limited Novelty: Reviewers (Euk2, yx5Z, bFZ2) consistently noted that the method is an incremental engineering combination of standard techniques (MMD, CL, VAE) without sufficient theoretical insight or domain-specific innovation for physiological signals.

Insufficient Evaluation & Baselines: Multiple reviewers (Euk2, yx5Z) pointed out the omission of strong, recent baselines (e.g., BSTT [ICLR 2023], SleepSMC [ICLR 2025]). Additionally, the comparison with SleepDG (a Domain Generalization method) in a UDA setting was flagged as unfair and methodologically unsound.

Methodological Clarity: There is a fundamental concern regarding the conflation of UDA and DG settings (yx5Z, bFZ2), and a lack of justification for why specific components (like the "Overview" structure or specific losses) are necessary (HBSE).

Reproducibility: The lack of code and implementation details prevents verification (Euk2).

**Reviewer Scores:**

As there appear to be no author responses or revisions visible that effectively resolved the major issues。 The reviewers would likely maintain their low scores. The concerns raised are fundamental to the paper's contribution and evaluation strategy (e.g., lack of novelty, missing modern baselines), which typically cannot be addressed during the rebuttal phase without a complete overhaul of the experiments and methodology.

---

### Decision · Program_Chairs · 2026-01-26

Reject